# Screening a library of antibacterial compounds leads to discovery of novel inhibitors for *Neisseria gonorrhoeae* and *Chlamydia trachomatis*

**Abdallah S. Abdelsattar**[1,2], **Nader S. Abutaleb**[1,2], **Mohamed N. Seleem**[1,2]*

**1** Department of Biomedical Sciences and Pathobiology, Virginia-Maryland College of Veterinary Medicine, Virginia Polytechnic Institute and State University, Blacksburg, Virginia, United States of America, **2** Center for One Health Research, Virginia Polytechnic Institute and State University, Blacksburg, Virginia, United States of America

* naguieb@vt.edu

## Abstract

*Chlamydia trachomatis* and *Neisseria gonorrhoeae* were the most common bacteria causing sexually transmitted infections (STIs) in 2020, with 211 million cases worldwide. Despite the fact that the co-infections of *N. gonorrhoeae* with *C. trachomatis* are common, there is no single treatment effective against both pathogens. Ceftriaxone, the current recommended drug for gonococcal infections, is not effective against *C. trachomatis*. Additionally, *N. gonorrhoeae* has developed resistance against the drugs recommended for treating chlamydial infections. Therefore, new drugs capable of treating *C. trachomatis*/*N. gonorrhoeae* co-infections are needed. Drug repurposing is an attractive, fast-track approach for antimicrobial drug discovery. In an attempt to address the unmet need for development of *C. trachomatis*/*N. gonorrhoeae* therapeutics utilizing the drug repurposing approach, we screened the antibacterial compounds library against *N. gonorrhoeae*. This library encompasses a unique collection of 1,128 bioactive compounds with validated antibacterial activities. A total of 172 active hits were identified, and then repeated drugs with different salts or previously reported drugs were excluded before determining the minimum inhibitory concentrations (MICs) against *N. gonorrhoeae* FA1090. Thereafter, the anti-*C. trachomatis* activities of the 14 selected drugs were assessed. We identified gloxazone and SPR719 as promising agents with potent activities against both *C. trachomatis* and *N. gonorrhoeae* (MICs ≤ 1 µM). Collectively, SPR719 and gloxazone could be considered promising agents warranting further investigation to address the unmet need in the treatment of bacterial STIs.

**Data availability statement:** All relevant data are within the manuscript and its Supporting Information files.

**Funding:** The author(s) received no specific funding for this work.

**Competing interests:** The authors have declared that no competing interests exist.

## 1. Introduction

Sexually transmitted infections (STIs) greatly impact sexual and reproductive health globally, potentially causing serious issues, including genital ulcers, endocarditis, maternal mortality, pelvic inflammatory disease, infertility, lower abdominal pain, arthritis, and discharges from the urethra or vagina [1–5]. More than 1 million curable STIs are acquired daily worldwide in people aged 15–49 years, the majority of which are asymptomatic [6]. Bacterial STIs remain a globally significant public health problem, accounting for millions of infections annually. *C. trachomatis* and *N. gonorrhoeae* are the most prevalent bacterial STIs, with an estimated number of new cases of 129 and 82.4 million, respectively, in 2020 [6–8]. Infections caused by these bacteria can result in several medical complications, including infertility, ectopic pregnancy, eye inflammation, trachoma, arthritis, heart diseases, and perinatal mortality [9–13].

*N. gonorrhoeae* has progressively developed resistance to nearly all FDA-approved therapeutics. According to the Centers for Disease Control (CDC), azithromycin was removed from the guidelines of treating gonococcal infections, leaving ceftriaxone, as a single injection, as the only recommended drug for treatment of gonorrhea with no effective oral drug available [14–16]. However, over the last decade, gonococcal strains with ceftriaxone and high-level azithromycin resistance have begun to appear worldwide [17–20]. The increasing prevalence of antimicrobial resistance (AMR) in *N. gonorrhoeae* has led to the emergence of gonorrhea superbugs (also known as super gonorrhea), which could result in untreatable gonococcal infections globally in the future [21,22]. Gonorrhea superbugs are extensively drug-resistant *N. gonorrhoeae* with high-level resistance to not only ceftriaxone, but also to other classes such as penicillins, sulphonamides, tetracyclines, fluoroquinolones and macrolides (including azithromycin) [23]. Hence, in 2024, the WHO listed fluoroquinolone-resistant *N. gonorrhoeae* as a high-priority microorganism, surpassing methicillin-resistant *Staphylococcus aureus* (MRSA) [24]. Therefore, discovering effective and safe anti-*N. gonorrhoeae* therapies are a high priority globally.

*C. trachomatis*, an obligate intracellular pathogen, is the most common bacterium that causes STIs [7,8]. The current recommended treatment for *C. trachomatis* infections is doxycycline, especially among adolescents and adults [25]. Other alternative treatments, azithromycin and levofloxacin, have been associated with reported treatment failures and the emergence of resistance [26–28]. Co-infection rates of chlamydia and gonorrhea are estimated between 50% and 70%, and there is no single treatment effective for both pathogens [15,29]. Treating *C. trachomatis*/*N. gonorrhoeae* co-infections is challenging because doxycycline, the standard-of-care for chlamydia is ineffective against gonorrhea [15], and ceftriaxone, the current last-line treatment for gonorrhea, is ineffective against chlamydia [30,31]. In light of the above, there is an urgent need to identify an oral antibiotic effective against both *C. trachomatis* and *N. gonorrhoeae*.

Despite recent technological advances, the de novo process of discovering a novel drugs remains a complex and time-consuming task that could last for 15 years and cost an average of $2–3 billion [32]. However, one of the most promising and rapid strategies to combat the escalating AMR crisis is the repurposing

of FDA-approved drugs. This approach has the potential of discovering new treatment options, offering a beacon of hope in the fight against gonorrhea [33]. The success of this strategy is evident in the repurposing of several drugs for antimicrobial use, such as doxycycline for malaria, amphotericin B and paromomycin for leishmaniasis [34–37], and spiramycin for toxoplasmosis [38]. The objective of the study was to conduct a rigorous screening of the antibacterial compounds library (HY-L049), which contains agents with confirmed bioactivity and safety, as demonstrated through preclinical and clinical trials, to identify potential agents against both *N. gonorrhoeae* and *C. trachomatis.* Additionally, the antibacterial activity of selected compounds was confirmed against *N. gonorrhoeae* and *C. trachomatis* strains*.*

## 2. Materials and methods

### 2.1. Bacterial strains and reagents

*N. gonorrhoeae* FA1090, *C. trachomatis* serovar L2 (ATCC VR-902B), and McCoy cell line were purchased from the American Type Culture Collection (ATCC). The media and supplements utilized in this work were purchased as follows: Phosphate-buffered saline (PBS) (Corning, NY, USA), nicotinamide adenine dinucleotide (NAD), pyridoxal, and hematin (Chem-Impex International, Wood Dale, IL, USA), and IsoVitaleX, brucella broth, chocolate II agar plates, and bovine hemoglobin, (Becton, Dickinson and Company, Cockeysville, MD, USA). Cycloheximide (Fisher Scientific, Fail Lawn, NJ, USA), *C. trachomatis* major outer membrane protein (MOMP) primary and secondary antibody (Bio-Rad, Hercules, CA, USA). Hoechst 33342 (MedChemExpress, Monmouth Junction, NJ, USA), azithromycin (TCI America, Portland, OR, USA), the phalloidin conjugates California red (AAT-Bioquest, Pleasanton, CA, USA) and Eagle's Minimum Essential Medium (EMEM) (Sigma-Aldrich, St. Louis, MO, USA).

### 2.2. The antibacterial compounds library

The antibacterial compounds library (HY-L049), which contains a unique collection of 1,128 bioactive antibacterial compounds, was purchased from MedChemExpress (Princeton, Monmouth Junction, NJ, USA). The library was supplied in 96-well plates of 10 mM stocks of the drugs and clinical molecules dissolved in either dimethyl sulfoxide (DMSO), ethanol, or water.

### 2.3. Screening assay

The antibacterial compounds library was screened against *N. gonorrhoeae* FA1090 at a fixed concentration of 1 µM to evaluate anti-gonococcal activity as described previously [39–42]. Briefly, the *N. gonorrhoeae* FA1090 was grown on a chocolate II agar plate supplemented with IsoVitaleX. Single colonies were suspended in PBS and adjusted to a turbidity equivalent to 1 McFarland standard. Next, bacteria were diluted in Brucella broth supplemented with 1% IsoVitaleX to achieve a titer of about $1 \times 10^6$ CFU/mL and added to 96-well plates containing the library drugs and clinical molecules (1 µM). Wells containing DMSO served as a growth control. The plates were incubated at 37°C in the presence of 5% $CO_2$ for 24 h. Thereafter, SpectraMax i3 multi-mode microplate reader (Molecular Devices, Sunnyvale, CA, USA) was utilized to measure the $OD_{600}$. The percentage of inhibition for each tested bioactive compound was calculated using the following equation.

$$\% \text{ inhibition} = \frac{\text{OD600 of bacteria incubated with tested agent}}{\text{OD600 of bacteria incubated with DMSO}} \times 100$$

Drugs and clinical molecules with an inhibition level of ≥90% were deemed as active hits, and their antibacterial activity was confirmed. GraphPad Prism 9.0 (Graph Pad Software, La Jolla, CA, USA) was utilized to illustrate the growth inhibition percentage.

### 2.4. Evaluation of MICs of the selected hits against *N. gonorrhoeae*

The selected hits were subsequently tested against *N. gonorrhoeae* FA1090 to confirm their MICs, using the broth micro-dilution method, as described previously [39–44]. Briefly, serial dilutions of the active hits and the control antibiotics (azithromycin and ceftriaxone) were incubated with bacterial solution (~ 1 × 10^6 CFU/mL) at 37 °C in the presence of 5% $CO_2$ for 24 h. The bacteria with DMSO served as a negative control. MICs were defined as the minimum concentrations of test agents that completely inhibited bacterial growth as determined by visual inspection of plates.

### 2.5. Investigation of the anti-chlamydial activity of selected hits

The anti-chlamydial activity of the selected hits was evaluated against *C. trachomatis* serovar L2 in McCoy cells, as previously described [45,46]. Briefly, the McCoy cell line was cultured in EMEM and incubated at 37°C with 5% $CO_2$. Thereafter, cells were infected with *C. trachomatis* L2 (multiplicity of infection (MOI) = 1), in the presence of cycloheximide (1 μg/mL) for 2 hours. Subsequently, cells were treated with the test agents (1 μM). Azithromycin (1μM) and medium without antibiotics served as positive and negative controls, respectively. After 48 hours of treatment, the cells were stained with Hoechst 33342 for the DNA, phalloidin conjugated to California Red for the actin, in addition to *C. trachomatis* MOMP antibody and donkey anti-goat IgG conjugated to Alexa 488 as an indication for chlamydial inclusion. Finally, the images were taken using a Nikon Eclipse Ti2 inverted microscope (Nikon, Melville, NY, USA).

## 3. Results and discussion

### 3.1. Screening the antibacterial compounds library against *N. gonorrhoeae*

The antibacterial compounds library (containing 1,128 drugs and clinical molecules) was screened against *N. gonorrhoeae* FA1090, at a concentration of 1 μM, using the broth microdilution assay. Although the Clinical and Laboratory Standards Institute (CLSI) recommends the agar dilution method for antimicrobial susceptibility testing of *N. gonorrhoeae,* broth microdilution is used to screen drug libraries as established in a previously reported work [33]. The advantages of broth microdilution include its reproducibility, minimal sample requirement, low cost, and the ability to evaluate a large number of compounds [47]. The library screening identified 172 unique drugs and clinical molecules that inhibited the growth of *N. gonorrhoeae* FA1090 by more than 90% at 1 μM (S1 Table and Fig 1). In addition, 22 repeated drugs with different salts were eliminated. After analyzing the identified hits, we found that 127 had previously been reported against *N. gonorrhoeae* (S1 Table) and were therefore excluded. In addition, topical agents (such as antiseptics, surfactants, and fluorochrome agents), toxins, and anti-cancer agents were excluded, leaving 14 agents as promising hits.

The 14 agents identified, which were not previously reported against *N. gonorrhoeae*, were screened against *N. gonorrhoeae* FA1090, and their MICs were determined. They displayed potent activity against *N. gonorrhoeae* FA1090 with MIC values ranging from 0.008 μM to 1 μM (Table 1).

Beta-lactam antibiotics, which inhibit penicillin-binding proteins, thereby preventing the synthesis of peptidoglycan, are widely used in the management and treatment of bacterial infections due to their potent and broad-spectrum activity [48]. Currently, ceftriaxone, a beta-lactam antibiotic, is the only recommended treatment for *N. gonorrhoeae* infections [15,49,50]. Beta-lactam antibiotics make up about 65% of the total antibiotics market [51]. This could explain why the highest number of active hits (47) identified in our screening against *N. gonorrhoeae* belongs to the beta-lactam antibiotic class. Among these 47 beta-lactam antibiotic hits, two antibiotics were not previously reported against *N. gonorrhoeae* (aspoxicillin and ceftiofur). Aspoxicillin (TA-058) is an old semisynthetic penicillin compound that has a broad spectrum of activity against both Gram-positive and Gram-negative bacteria [52]. Aspoxicillin exhibited potent activity against *N. gonorrhoeae* FA1090 (MIC = 0.5 μM). Ceftiofur is FDA-approved for the treatment of mastitis in lactating dairy cattle by intramammary infusion [53]. Although ceftiofur has a lower MIC of 0.008 μM than aspoxicillin (0.5 μM), previous reports have demonstrated that the ability of Gram-negative bacteria to develop resistance against ceftiofur [54].

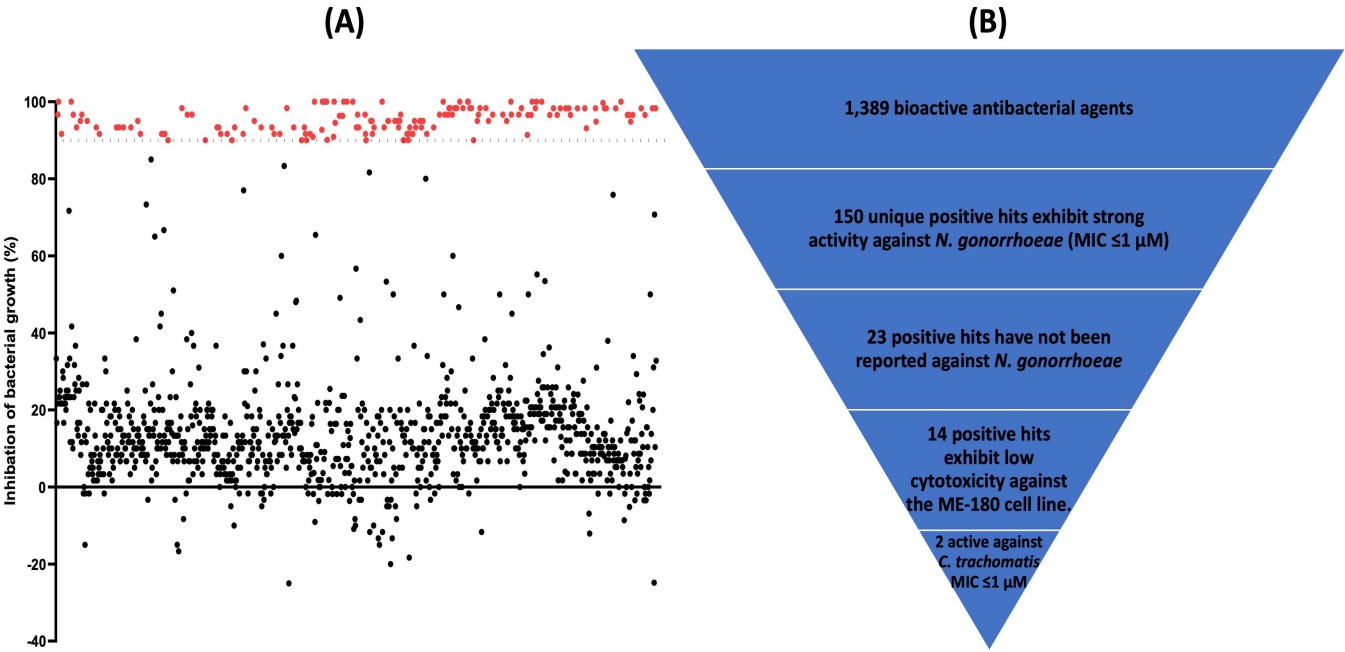

**Fig 1. Screening the antibacterial compounds library against *N. gonorrhoeae* FA1090. (A)** Results of screening the antibacterial compounds library (containing 1,128 bioactive antibacterial agents), at a fixed concentration of 1 µM, against *N. gonorrhoeae* FA1090. Drugs exhibiting ≥ 90% inhibition of bacterial growth were considered positive hits (red). **(B)** Schematic of the screening process in this study, and the number of test agents at each step to finally obtain two molecules that are potent against *N. gonorrhoeae* and *C. trachomatis*.

**Table 1. MIC values of the selected hits against *N. gonorrhoeae* FA1090.**

| Drug name | MIC (µM) | The potential mechanism of action |
|---|---|---|
| Aspoxicillin | 0.5 | Inhibits cell wall synthesis |
| Ceftiofur | 0.008 | Inhibits cell wall synthesis |
| Azathramycin | 1 | Inhibits protein synthesis |
| Virginiamycin M1 | 0.125 | Inhibits protein synthesis |
| Marbofloxacin | 0.064 | Inhibits DNA synthesis |
| Orbifloxacin | 0.032 | Inhibits DNA synthesis |
| Levonadifloxacin | 0.032 | Inhibits DNA synthesis |
| Fluoroquinolonic acid | 0.064 | Inhibits DNA synthesis |
| SPR719 | 0.125 | Inhibits DNA synthesis |
| Furaltadone | 1 | Inhibits the synthesis of DNA, RNA, and protein |
| Nifursol | 1 | Inhibits the synthesis of DNA, RNA, and protein |
| Nifurpirinol | 1 | Inhibits the synthesis of DNA, RNA, and protein |
| Antibacterial agent 18 | 1 | hinders transpeptidation and glycosylation |
| Gloxazone | 0.064 | Unknown |

Macrolides, streptogramins, and phenicols are antibiotics that inhibit bacterial protein synthesis by binding to the bacterial 50S ribosomal subunit [55,56]. The macrolide antibiotic azithromycin was recommended in combination with ceftriaxone as the standard of care for the treatment of gonococcal infections until 2020 [16]. In addition, the CDC recommends a combination of azithromycin and gentamicin as an alternative treatment for uncomplicated gonococcal infection of the

cervix, urethra, or rectum if a cephalosporin allergy is identified [15]. A total of 14 agents belonging to macrolides, strepto-gramins, and phenicols were identified as hits from our screening. Among these hits, two antibiotics were not previously reported against *N. gonorrhoeae* (azathramycin (desmethyl azithromycin) and virginiamycin M1). The MIC of azathra-mycin (MIC = 1 μM) was less effective than that of azithromycin (MIC = 0.25 μM) against the FA1090 strain. Furthermore, virginiamycin M1, a polyunsaturated macrocyclic lactone produced by *Streptomyces virginiae*, belonging to the strepto-gramin A group [57], exhibited a potent activity similar to that of azithromycin against *N. gonorrhoeae* FA 1090. Virginiamy-cin is widely used as a growth promoter and for the prevention and treatment of infections in farm animals.

The antibacterial activity of pleuromutilin derivatives is mediated by inhibiting bacterial protein synthesis after binding to the bacterial 50S ribosomal subunit [58,59], similar to azithromycin. However, earlier reports have indicated a slight dif-ference in activity against azithromycin-resistant bacteria due to differences in binding sites [60–62]. Recently, lefamulin, a pleuromutilin derivative, has been assessed as a potential drug against STIs, including *N. gonorrhoeae and Chlamydia trachomatis* [63].

Recently, greater attention has been given to drugs that inhibit DNA synthesis in *N. gonorrhoeae,* such as zoliflodacin and gepotidacin [64]. Additionally, ciprofloxacin was one of the recommended treatments for *N. gonorrhoeae* until 2007 [65]. Among the 14 non-previously reported identified hits, four drugs belonging to the fluoroquinolones were identified (marbofloxacin, orbifloxacin, levonadifloxacin, and fluoroquinolonic acid), which showed an MIC range of 0.032–0.064 μM against *N. gonorrhoeae* FA1090. Marbofloxacin and orbifloxacin were approved by the FDA for the treatment of infections in dogs and cats [66,67]. Recently, levonadifloxacin was approved in India as an oral and injectable drug for the treatment of acute bacterial skin and skin structure infections [68,69]. In addition, fluoroquinolonic acid was identified as a ciprofloxa-cin impurity with inhibitory effect against *E. coli* MG1655 at a low concentration of 0.064 μg/mL [70].

Nitrofurans are a class of nitroaromatic antibacterial agents that require activation inside bacterial cells to form reactive intermediates that damage macromolecules. This reactivation is performed by the bacterial flavoprotein nitrofuran reduc-tase which reduces the nitrofuran antibiotic and its derivatives to a reactive form, which then inhibits the synthesis of DNA, RNA, and proteins [71]. The analysis of the positive hists revealed 6 active hits from the nitrofurans class, 3 of which were not previously investigated against *N. gonorrhoeae* (furaltadone, nifursol, and nifurpirinol). They displayed potent activity against *N. gonorrhoeae* with MIC values of 1 μM (Table 1).

In addition to the previously identified hits, we identified other 5 active drugs and clinical molecules which were not reported before against *N. gonorrhoeae* (nigericin, SPR719, antibacterial agent 18, walrycin B, and gloxazone), with MIC values ranging from 0.06 to 1 μM. Nigericin, a polyether ionophore antibiotic, which was reported to have bactericidal activity against both Gram-positive and Gram-negative bacteria by disrupting the ionic balance [72,73]. Nigericin inhibited *N. gonorrhoeae* FA1090 at an MIC value of 0.5 μM. Additionally, SPR719 a novel benzimidazole antibiotic that inhibits the ATPase activity of the DNA gyrase (GyrB) and topoisomerase IV (ParE) in mycobacteria [74]. SPR720 is in clinical development for the treatment of non-tuberculous mycobacterial pulmonary disease (NTM-PD) and pulmonary tuberculo-sis [75]. SPR719 and its prodrug, SPR720, showed remarkable safety, tolerability, and pharmacokinetics results in clinical trials (NCT03796910) [76]. SPR719 demonstrated a potent anti-gonococcal activity, inhibiting the tested *N. gonorrhoeae* strain at a concentration of 0.125 μM. Furthermore, antibacterial agent 18 was previously reported as an active compound against Gram-negative bacterial pathogens, with an MIC ranging from 0.125 to 1 μg/mL, by disrupting their cell wall syn-thesis via hindering transpeptidation and glycosylation [77]. Antibacterial agent 18 showed an MIC value of 1 μM against *N. gonorrhoeae* FA1090. Additionally, the analysis of the positive hits revealed a potent activity of walrycin B against *N. gonorrhoeae* FA1090 (MIC = 0.5 μM). Walrycin B is an antibacterial compound that has been shown to target the WalK/WalR two-component signal transduction system, which is essential for the viability of many Gram-positive bacteria. This inhibition leads to the inhibition of crucial genes required for cell wall synthesis and coordinating cell division [78]. Further-more, gloxazone, an effective drug against anaplasmosis [79], exhibited potent activity against *N. gonorrhoeae* FA1090 (MIC = 0.06 μM) (Table 1).

### 3.2. Activity of the selected positive hits against *C. trachomatis*

Co-infection rates of chlamydia and gonorrhea are high. Women with gonorrhea are co-infected with chlamydia in 17.6% – 57.9% of cases, while women with chlamydia are co-infected with gonorrhea in 2.1% – 17.2% of cases [30,31,80,81]. Moreover, during the co-infection, *N. gonorrhoeae* can induce *C. trachomatis* to enter a difficult-to-treat persistence-like state [80]. Since treating *N. gonorrhoeae/C. trachomatis* co-infections are challenging because there is no available drug effective against both pathogens. It was also reported that treatment with non-specific antibiotics can increase rates of *N. gonorrhoeae* resistance [82,83]. Hence, the search for an oral antibiotic targeting both pathogens cannot be overemphasized. Therefore, we sought to investigate the activity of the selected hits from our screening against *C. trachomatis* L2. Agents with reported toxicity, including walrycin B [84], and Nigericin [85] were excluded. We have also excluded the active hits belonging to known classes of antibiotics (beta-lactams, macrolides, fluoroquinolones, and nitrofurans). We selected only 2 hits (SPR719, and gloxazone) to evaluate their activity against *C. trachomatis*. We screened their activity, at a fixed concentration of 1 µM, against *C. trachomatis* L2 infecting McCoy cells. SPR719 and gloxazone displayed potent activity against *C. trachomatis*, completely clearing the bacteria at the tested concentration (MIC ≤ 1 µM) (Fig 2).

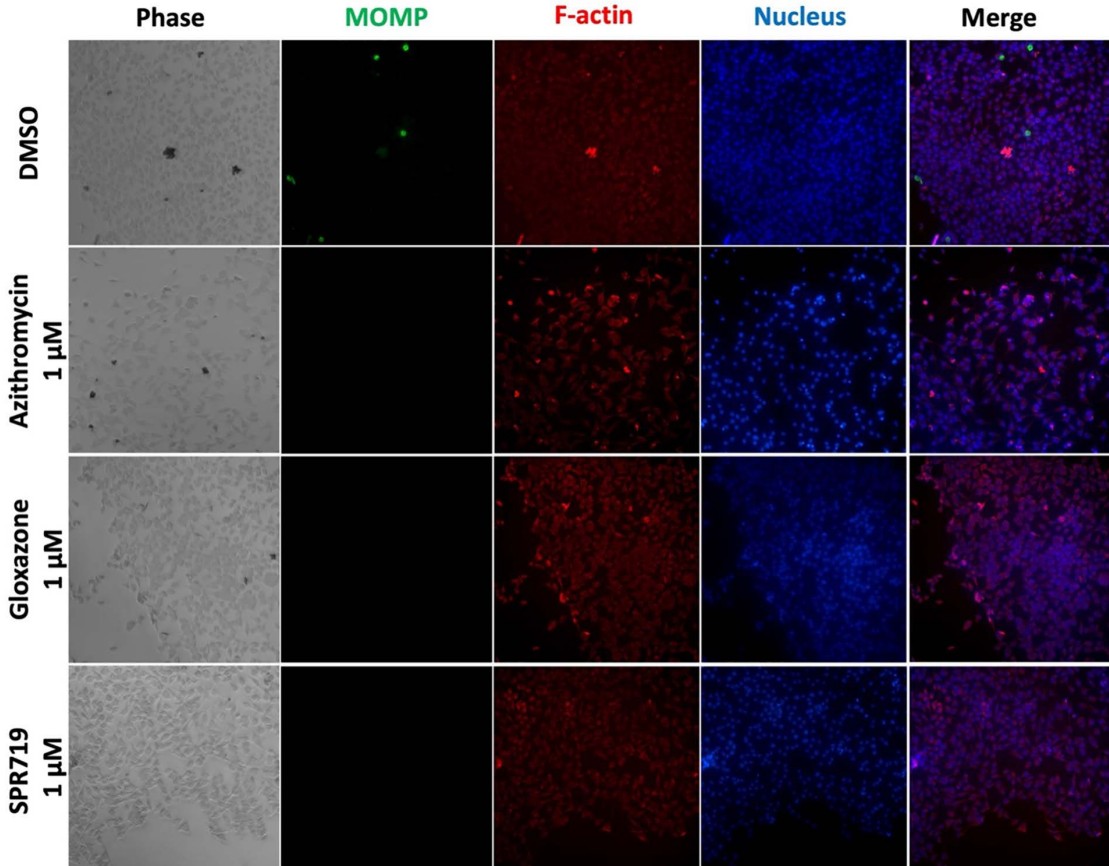

**Fig 2. Immunofluorescence images showing the inhibitory effects of SPR719 and gloxazone (at a fixed concentration of 1 µM).** McCoy cells were infected and incubated for 2 hours before being exposed to SPR719 and gloxazone (at 1 µM), and incubated for 48 hours at 37°C with 5% $CO_2$. Azithromycin at 1 µM was used as a positive control. The untreated cells (DMSO) served as a negative control. Nuclear DNA was stained with Hoechst 33342 (blue), primary antibodies bound to the MOMPs of *Chlamydia* were stained with donkey anti-goat IgG conjugated to Alexa 488 (green), and phalloidin conjugated to California red was used to stain F-actin (red).

The addition of SPR719, gloxazone, or azithromycin at a fixed concentration (1 μM) completely blocked the development of *C. trachomatis* inclusion bodies. However, the negative control (DMSO) shows the mature inclusions after 48 hours. Inclusion bodies were identified using primary antibodies bound to *Chlamydia* MOMPs and a secondary antibody conjugated to Alexa 488, which appears as green circles (Fig 2). We recommend further in vitro studies to assess the inhibitory effects of hit compounds (gloxazone and SPR719) on both *N. gonorrhoeae* and *C. trachomatis* simultaneously in a co-culture assay [86].

## Conclusion

In conclusion, this study aims to identify new therapeutics for *N. gonorrhoeae* and *C. trachomatis* through repurposing existing drugs and compounds in clinical trials. Screening a library of antibacterial compounds led to the discovery of novel lead molecules, such as SPR719 and gloxazone, that possess potent in vitro anti-*N. gonorrhoeae* activity. They also showed potent activity against *C. trachomatis.* These agents warrant further investigation for the development of new anti-*C. trachomatis*/*N. gonorrhoeae* therapeutics.

## Supporting information

**S1 Table. The positive hit compounds against *N. gonorrhoeae* FA1090.**
(DOCX)

## Acknowledgments

We gratefully acknowledge Dr. Xiaogang Wang for providing the *C. trachomatis* serovar L2.

## Author contributions

**Conceptualization:** Mohamed N. Seleem.

**Formal analysis:** abdallah S. abdelsattar, Nader S. Abutaleb.

**Investigation:** Mohamed N. Seleem.

**Methodology:** abdallah S. abdelsattar, Nader S. Abutaleb.

**Supervision:** Mohamed N. Seleem.

**Writing – original draft:** abdallah S. abdelsattar.

**Writing – review & editing:** Nader S. Abutaleb, Mohamed N. Seleem.

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
