## [Decision Letter · Decision Letter 0]

12 Dec 2025

Dear Dr. Seleem,

We look forward to receiving your revised manuscript.

Kind regards,

Yung-Fu Chang

Academic Editor

PLOS One

Journal Requirements:

2. Please include captions for your Supporting Information files at the end of your manuscript, and update any in-text citations to match accordingly. Please see our Supporting Information guidelines for more information: http://journals.plos.org/plosone/s/supporting-information .

Additional Editor Comments:

Your manuscript has been reviewed by an expert in your field. The comments are enclosed for your references. Based on the comments that your manuscript needs a minor revision before a decision can be made.

Reviewer's Responses to Questions

**Comments to the Author**

1. Is the manuscript technically sound, and do the data support the conclusions?

Reviewer #1: Yes

2. Has the statistical analysis been performed appropriately and rigorously?

Reviewer #1: Yes

3. Have the authors made all data underlying the findings in their manuscript fully available?

Reviewer #1: Yes

4. Is the manuscript presented in an intelligible fashion and written in standard English?

Reviewer #1: Yes

Reviewer #1: This manuscript effectively addresses the critical unmet need for a single therapeutic agent against co-infections of Chlamydia trachomatis and Neisseria gonorrhoeae. Utilizing a drug repurposing approach, the authors screened an antibacterial compound library and successfully identified gloxazone and SPR719 as promising dual-action agents, demonstrating potent activity against both pathogens. The findings are highly significant as they propose immediate candidates for further preclinical development to combat the growing crisis of antibiotic-resistant sexually transmitted infections (STIs). Given the quality of the data and the clinical importance of the findings, the manuscript is well-suited for publication with minor revisions. My detailed comments are below:

Abstract: I assume the 211 million is the number of cases worldwide. Please specify it.

Method: In the section describing the MIC determination against N. gonorrhoeae, please specify what broth medium was used.

Result:

1. The first paragraph of the results section doesn’t fit in there because it only mentions anti- N. gonorrhoeae therapeutics. Please consider deleting it or re-writing it.

2. The description of the Immunofluorescence images is insufficient. It is currently difficult to interpret the outcome of the compound treatment from the images alone. Please add a more detailed description explaining what the key markers/colors represent and how the compound treatment visibly alters the morphology or life cycle of C. trachomatis. Also, please clarify if a series of concentrations was tested for the IF images to correlate with the MIC determination.

3. Given that N. gonorrhoeae is known to potentially induce a difficult-to-treat persistence-like state in C. trachomatis, it is highly relevant to the goal of finding a single co-infection therapeutic. Have you tested the inhibitory effects of your hit compounds (gloxazone and SPR719) on both N. gonorrhoeae and C. trachomatis simultaneously in a co-culture assay? If not, this should be discussed as a future direction.

Reference: Please ensure that the format of the references is consistent throughout the entire Reference section.

**Do you want your identity to be public for this peer review?** For information about this choice, including consent withdrawal, please see our Privacy Policy

Reviewer #1: No

---

## [Author Response · Author response to Decision Letter 1]

15 Dec 2025

Reviewer #1:

This manuscript effectively addresses the critical unmet need for a single therapeutic agent against co-infections of Chlamydia trachomatis and Neisseria gonorrhoeae. Utilizing a drug repurposing approach, the authors screened an antibacterial compound library and successfully identified gloxazone and SPR719 as promising dual-action agents, demonstrating potent activity against both pathogens. The findings are highly significant as they propose immediate candidates for further preclinical development to combat the growing crisis of antibiotic-resistant sexually transmitted infections (STIs). Given the quality of the data and the clinical importance of the findings, the manuscript is well-suited for publication with minor revisions. My detailed comments are below:

1. Abstract: I assume the 211 million is the number of cases worldwide. Please specify it.

1. Response: Thank you for this comment. Yes, it is worldwide, so we added the word “worldwide” to specify the number.

2. Method: In the section describing the MIC determination against N. gonorrhoeae, please specify what broth medium was used.

2. Response: We added that the media was “Brucella broth supplemented with 1% IsoVitaleX.”

3. Result: 1. The first paragraph of the results section doesn’t fit in there because it only mentions anti- N. gonorrhoeae therapeutics. Please consider deleting it or re-writing it.

3. Response: Based on your recommendation, we deleted the first paragraph of the results section.

4. Result: 2. The description of the Immunofluorescence images is insufficient. It is currently difficult to interpret the outcome of the compound treatment from the images alone. Please add a more detailed description explaining what the key markers/colors represent and how the compound treatment visibly alters the morphology or life cycle of C. trachomatis. Also, please clarify if a series of concentrations was tested for the IF images to correlate with the MIC determination.

4. Response: Thank you. We added a paragraph in the results section and modified the legend to explain the color code. “The addition of SPR719, gloxazone, or azithromycin at a fixed concentration (1 μM) completely blocked the development of C. trachomatis inclusion bodies. However, the negative control (DMSO) shows the mature inclusions after 48 hours. Inclusion bodies were identified using primary antibodies bound to Chlamydia MOMPs and a secondary antibody conjugated to Alexa 488, which appears as green circles (Fig. 2). ”

5. Result: 3. Given that N. gonorrhoeae is known to potentially induce a difficult-to-treat persistence-like state in C. trachomatis, it is highly relevant to the goal of finding a single co-infection therapeutic. Have you tested the inhibitory effects of your hit compounds (gloxazone and SPR719) on both N. gonorrhoeae and C. trachomatis simultaneously in a co-culture assay? If not, this should be discussed as a future direction.

5. Response: Thank you for this comment, and we added a future direction in the last paragraph: “We recommend further in vitro studies to assess the inhibitory effects of hit compounds (gloxazone and SPR719) on both N. gonorrhoeae and C. trachomatis simultaneously in a co-culture assay.”

6. Reference: Please ensure that the format of the references is consistent throughout the entire Reference section.

6. Response: We confirmed the references are consistent.

---

## [Editor Report · Decision Letter 1]

22 Dec 2025

Screening a library of antibacterial compounds leads to discovery of novel inhibitors for Neisseria gonorrhoeae and Chlamydia trachomatis

PONE-D-25-61453R1

Dear Dr. ,Seleem

We’re pleased to inform you that your manuscript has been judged scientifically suitable for publication and will be formally accepted for publication once it meets all outstanding technical requirements.

Kind regards,

Yung-Fu Chang

Academic Editor

PLOS One
---

## [Editor Report · Acceptance letter]

PONE-D-25-61453R1

PLOS One

Dear Dr. Seleem,

I'm pleased to inform you that your manuscript has been deemed suitable for publication in PLOS One. Congratulations! Your manuscript is now being handed over to our production team.

Kind regards,

on behalf of

Dr. Yung-Fu Chang

Academic Editor

PLOS One